# Rebuilding core abscisic acid signaling pathways of *Arabidopsis* in yeast

Moritz Ruschhaupt[1] , Julia Mergner[2], Stefanie Mucha[1] , Michael Papacek[1], Isabel Doch[1],
Stefanie V Tischer[1], Daniel Hemmler[3,4] , David Chiasson[5] , Kai H Edel[6], Jörg Kudla[6],
Philippe Schmitt-Kopplin[3,4], Bernhard Kuster[2,7] & Erwin Grill[1,*]

## Abstract

The phytohormone abscisic acid (ABA) regulates plant responses to abiotic stress, such as drought and high osmotic conditions. The multitude of functionally redundant components involved in ABA signaling poses a major challenge for elucidating individual contributions to the response selectivity and sensitivity of the pathway. Here, we reconstructed single ABA signaling pathways in yeast for combinatorial analysis of ABA receptors and coreceptors, downstream-acting SnRK2 protein kinases, and transcription factors. The analysis shows that some ABA receptors stimulate the pathway even in the absence of ABA and that SnRK2s are major determinants of ABA responsiveness by differing in the ligand-dependent control. Five SnRK2s, including SnRK2.4 known to be active under osmotic stress in plants, activated ABA-responsive transcription factors and were regulated by ABA receptor complexes in yeast. In the plant tissue, SnRK2.4 and ABA receptors competed for coreceptor interaction in an ABA-dependent manner consistent with a tight integration of SnRK2.4 into the ABA signaling pathway. The study establishes the suitability of the yeast system for the dissection of core signaling cascades and opens up future avenues of research on ligand-receptor regulation.

**Keywords** abscisic acid; ABF; abiotic stress; clade A PP2C; RCAR/PYL/PYR1; SnRK2
**Subject Categories** Plant Biology; Signal Transduction
**The EMBO Journal (2019) 38: e101859**

See also: **G Dubeaux & JI Schroeder** (September 2019)

## Introduction

Plants are cost-effective bioreactors that require sunlight, minerals, water, and air for producing food, feed, and fiber. The bioengineering of plants requires regulatory switches for the control of functional modules. Phytohormones and their associated signaling pathways are paradigms for such switches by mediating signal-specific responses with high sensitivity and efficiency. Rebuilding such a regulatory unit in an organism devoid of it, e.g., a species from a different kingdom, offers the possibility to examine the functionality and sufficiency of the signaling components. For phytohormones, such efforts have been undertaken to study the transcriptional regulation relayed by auxin (Pierre-Jerome *et al*, 2014) and for analyzing ion channel control mediated by abscisic acid (ABA) in yeast and *Xenopus*, respectively (Geiger *et al*, 2011; Brandt *et al*, 2012).

Abscisic acid regulates water relations and growth in addition to abiotic stress adaptations associated with massive changes in gene expression (Cutler *et al*, 2010; Finkelstein, 2013). ABA responses can be separated in a rapid and slow response. The rapid response includes the gating of ion channels (Schroeder *et al*, 2001; Maierhofer *et al*, 2014), while the slow response is mediated by altered gene expression (Song *et al*, 2016). The core ABA signaling cascade comprises a three-step regulatory process involving the receptor complex, protein kinases as mediators, and targets such as ion channels and transcription factors. The regulatory component of ABA receptor (RCAR)/pyrabactin resistance 1 (PYR1)/PYR1-like (PYL) and a protein phosphatase 2C (PP2C) of clade A form the functional receptor complex, which inhibits downstream-acting SnF1-related kinase 2 (SnRK2; Ma *et al*, 2009; Park *et al*, 2009; Cutler *et al*, 2010; Raghavendra *et al*, 2010).

The complexity of ABA signal relay is a function of the number of signaling components. In *Arabidopsis*, there are 14 ABA-binding RCARs and nine PP2Cs with known coreceptor function (Fuchs *et al*, 2014; Tischer *et al*, 2017). In addition, at least three SnRK2s and five basic leucine zipper-type ABA-responsive transcription factors are involved in the nuclear ABA pathway (Finkelstein & Lynch, 2000; Fujita *et al*, 2009; Yoshida *et al*, 2015a). The ABA-responsive transcriptional regulators comprise ABI5, acting primarily during seed

1   Chair of Botany, TUM School of Life Sciences Weihenstephan, Technical University Munich, Freising, Germany
2   Chair of Proteomics and Bioanalytics, TUM School of Life Sciences Weihenstephan, Technical University Munich, Freising, Germany
3   Research Unit Analytical BioGeoChemistry (BGC), German Research Center for Environmental Health, Helmholtz Zentrum München, Neuherberg, Germany
4   Chair of Analytical Food Chemistry, TUM School of Life Sciences Weihenstephan, Technical University Munich, Freising, Germany
5   Faculty of Biology, Institute of Genetics, Ludwig Maximilian University of Munich, Munich, Germany
6   Institut für Biologie und Biotechnologie der Pflanzen, Universität Münster, Münster, Germany
7   Bavarian Center for Biomolecular Mass Spectrometry (BayBioMS), Technical University Munich, Freising, Germany
    *Corresponding author. Tel: +49-8161-5433; E-mail: erwin.grill@wzw.tum.de

germination, ABF1, ABF2/AREB1, ABF3/DPBF5, and ABF4/AREB2 (Yoshida *et al*, 2010, 2015a). The ABFs target the ABA-responsive regulatory *cis*-element (ABRE) and coordinate drought stress and ABA responses. ABFs are also activated by calcium-dependent protein kinases such as CPK4 (Zhu *et al*, 2007) contributing to the integration of other signaling pathways into ABA responses (Edel & Kudla, 2016).

The ABA receptors RCAR/PYL are grouped into three subfamilies, which differ in ABA sensitivities at basal ABA levels but overall show little selectivity in PP2C binding (Tischer *et al*, 2017). The ABA receptor interaction with PP2C has features of a pseudosubstrate-enzyme binding (Soon *et al*, 2012). In the presence of ABA, the interaction between RCAR and PP2C is stabilized, leading to inhibition of the phosphatase activity and allowing downstream signaling by the protein kinases OPEN STOMATA 1 (OST1; Belin *et al*, 2006; Ng *et al*, 2011; Soon *et al*, 2012; Xie *et al*, 2012). OST1 belongs to the protein kinase family of SnRK2s comprising 10 members of three subgroups (Kobayashi *et al*, 2004) in which the subgroup III members OST1/SnRK2.6/SRK2E, SnRK2.2/SRK2D, and SnRK2.3/SRK2I are key regulators of the ABA signaling pathway and growth control (Fujii *et al*, 2007, 2011; Fujita *et al*, 2009; Umezawa *et al*, 2009; Yoshida *et al*, 2015b, 2019).

Activation of OST1 occurs in the presence of ABA and RCAR by the release of OST1 from PP2C inhibition and subsequent auto-phosphorylation of the SnRK2 activation loop, while SnRK2.2 and SnRK2.3 require activation in trans (Boudsocq *et al*, 2007; Ng *et al*, 2011; Cai *et al*, 2014; Zhu, 2016). SnRK2s are evolutionary conserved and play prominent roles in abiotic stress responses involving salt and hyperosmotic stress (Lind *et al*, 2015; Sussmilch *et al*, 2017; Shinozawa *et al*, 2019). In *Arabidopsis*, subgroup I members SnRK2.1, SnRK2.4, SnRK2.5, and SnRK2.10 were specifically activated under hyperosmotic stress and emerged as regulators of hyperosmotic stress adaptation and ROS homeostasis (Boudsocq *et al*, 2004; McLoughlin *et al*, 2012; Krzywińska *et al*, 2016a; Soma *et al*, 2017; Szymańska *et al*, 2019). The subgroup II, SnRK2.7 and SnRK2.8, were found to be moderately activated by ABA signaling and play a possible role in osmotic stress responses (Mizoguchi *et al*, 2010). Analysis of *Arabidopsis* mutants deficient in multiple SnRK2s indicated that subgroup III members act together with SnRK2s of subgroups I and II independently of ABA in the osmotic stress response pathway (Boudsocq *et al*, 2007; Fujii & Zhu, 2009; Fujii *et al*, 2011). A negative feedback of ABA signaling onto osmotic stress responses was inferred by the hyper-activation of SnRK2s in an *Arabidopsis* mutant deficient in multiple ABA receptors (Zhao *et al*, 2018). Osmotic stress is known to trigger ABA-independent and ABA-dependent signaling pathways (Yoshida *et al*, 2014). The integration of the osmotic stress response into the ABA signal pathway is still not clear on a molecular level.

The core ABA signal pathway is well understood. However, the multitude of signaling components and their functional redundancy pose major obstacles in elucidating the roles of individual components within the relay system. Analyzing multiple knockout mutants is one approach to detangle functional redundancy; however, those mutants are frequently impaired in their physiology (Fujita *et al*, 2009; Fujii *et al*, 2011; Zhao *et al*, 2018) and it is challenging to distinguish primary from secondary effects. We decided to circumvent the functional redundancy encountered in plants by studying the combinatorics of ABA signaling components in yeast using wild-type *Arabidopsis* proteins. Three prominent unresolved questions were addressed as follows: Do different SnRK2s selectively target ABFs, are SnRK2s from subgroups I and II involved in ABA signaling, and whether SnRK2s differentially contribute to the ABA sensitivity of the signaling cascade.

# Results

### Functionality of the ABA signaling pathway in yeast

The high number of ABA signaling components with redundant functions has hampered the analysis of their regulation and specificity in activating distinct transcription factors. In *Arabidopsis*, key transcriptional regulators are the bZIP factors ABF1, ABF2/AREB1, ABF3, ABF4/AREB2, and ABI5 (Yoshida *et al*, 2010, 2015a), which might be differentially activated by SnRK2s. To test this possibility, we introduced a reporter gene in yeast, which is specifically activated by the ABA-responsive transcription factors. ABFs bind to the ABRE sequence *TACGTGGC* (Choi *et al*, 2000), of which four copies were positioned upstream of a minimal promoter controlling luciferase expression and the reporter construct was stably integrated into the yeast genome (Fig EV1A, Appendix Table S1). Functionality of the reporter was shown by ABRE-dependent luciferase induction in the presence of ABF1, ABF2, ABF3, and ABF4 fused to the yeast GAL4 activation domain (Fig EV1B). The activity of ABF transcription factors is known to be stimulated by their aminoterminal phosphorylation via SnRK2s (Furihata *et al*, 2006). Accordingly, ABFs without a GAL4 activation domain were strongly activated by co-expression of the protein kinase OST1 in yeast whereas in the absence of OST1 basal levels of reporter gene expression (ABF3) or relatively weak induction (four-fold for ABF2) were observed (Fig 1A). In the presence of OST1, ABF4 conferred the highest reporter activation of more than 500-fold, followed by ABF2, while ABF1 and ABF3 caused low but significant reporter induction (Fig 1A). The OST1-mediated ABF2 activation required a functional protein kinase (Fig EV1C), and galactose-regulated OST1 expression revealed optimum induction levels 16–24 h after galactose administration (Fig EV1D–G). OST1-induced reporter activation was abrogated by co-expression of the PP2C ABI1 but not by a catalytically inactive variant (Fig EV2). The PP2C-dependent regulation was protein kinase-specific since the ABF2 activation by another protein kinase, the calcium-dependent CPK4 (Zhu *et al*, 2007), was not significantly inhibited by ABI1 (Fig EV2).

The expression of ABF2, OST1, and ABI1 together with the ABA receptor RCAR11 assembled the components required for a core ABA signaling pathway in yeast (Fig 1B, Appendix Fig S1). Exogenously applied ABA is taken up by yeast (Park *et al*, 2009), and in the presence of 0.1 mM ABA, the ABI1-mediated inhibition of the signal pathway was alleviated in the presence of RCAR11. The ABA- and ABA receptor-dependent reporter expression confirmed the functionality of the ABA signal cascade in yeast.

### ABA-response regulation by different ABA receptor complexes

The functionality of the yeast system prompted us to examine the ABA receptors. Analysis in plant protoplasts revealed different ABA-response sensitivities mediated by the ABA receptors (Tischer *et al*,

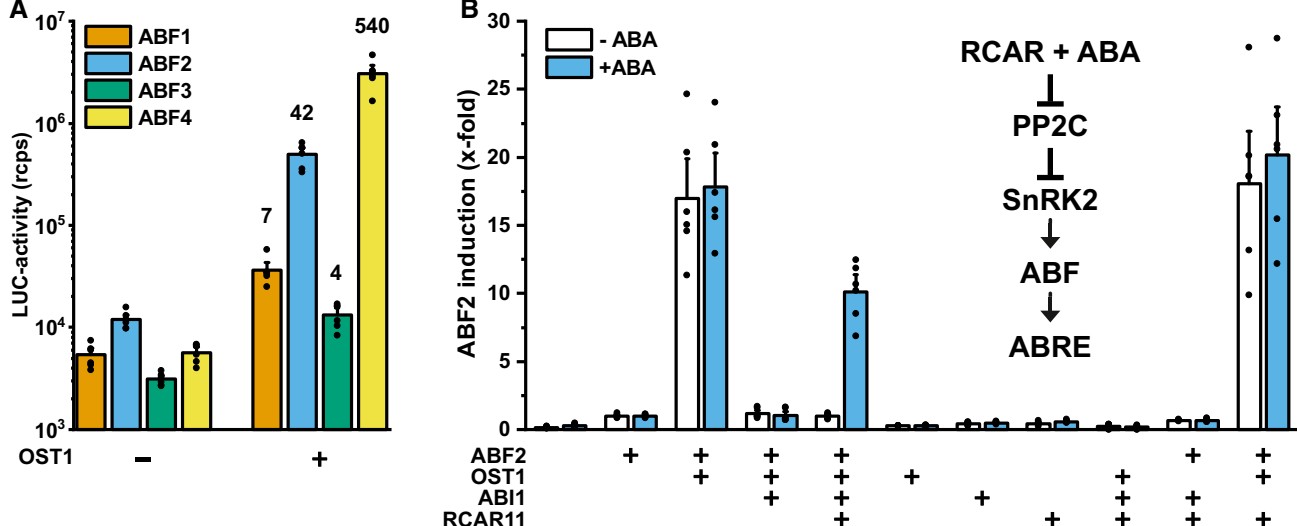

Figure 1.  Reconstruction of the ABA signal pathway in yeast.

A  The SnRK2 OST1 activates ABFs for the expression of a luciferase reporter (LUC) in yeast. Luciferase activity is given on a $\log_{10}$ scale as relative counts of light emission per second (rcps) normalized to the cell density. The OST1-mediated induction factor is indicated above the columns. In the absence of ABFs, the LUC activity was $3 \times 10^3$ rcps.

B  Combining the ABA signaling components ABF2, OST1, the PP2C ABI1, and RCAR11 provides ABA-controlled LUC expression. The reporter is under the control of a minimal promoter that contains ABA-responsive *cis* elements (ABREs). Core ABA signaling involves the RCAR receptor that inhibits the ABA coreceptor (PP2C) in the presence of ABA and allows the activation of the protein kinase SnRK2. SnRK2 phosphorylates and activates the key transcription factor ABF, which targets promoters containing ABRE. Reporter activity was determined 16 h after exogenous ABA (0.1 mM) administration and referred to samples expressing ABF2 alone.

Data information: The bar-plot with single data points represents the mean ± SEM; $n = 6$ biological replicates derived from three independent yeast transformants.
Source data are available online for this figure.

2017), which might be recapitulated in yeast. All 14 *Arabidopsis* RCARs were expressed individually in yeast together with ABF2, OST1, and ABI1 in the presence and absence of exogenous ABA and compared to the ABA-response regulation achieved by ectopic expression of the receptors in *Arabidopsis* protoplasts (Fig 2). The ABA response was assessed in plant protoplasts by induction of an ABF-responsive luciferase reporter. In yeast, ABA enhanced the reporter activity in the presence of ABA receptors except for RCAR4 and RCAR7 (Fig 2A; $P < 0.001$; one-tailed *t*-test; see Appendix Table S2). Without ABA, the subfamily I receptors RCAR1, RCAR3, and RCAR4 activated signaling in yeast significantly ($P < 0.001$; one-tailed *t*-test; see Appendix Table S2). These results were to a large extent comparable to the analysis in *Arabidopsis* protoplasts (Fig 2B). Activation of the ABA response by subfamily III RCARs required exogenous ABA administration both in yeast and in protoplasts. In the absence of exogenous ABA, several subfamily II RCARs showed minor activation in protoplasts with approximately 20 nM endogenous ABA (Tischer *et al*, 2017) but not in yeast. Taken together, the comparison of the response regulation by ABA receptors in yeast and *Arabidopsis* protoplasts revealed a similar tendency despite expected differences in the expression and intracellular regulation of ABA signaling components in plant and fungal cells.

For a more detailed analysis, combinations of the receptors RCAR12 and RCAR13 with the coreceptors ABI1, ABI2, and HAB1 were chosen for comparative ABA titration analyses in yeast, protoplasts, and *in vitro* (Fig 3). The receptor components were either expressed in yeast together with OST1 and ABF2, co-expressed in

*Arabidopsis* protoplasts for RCAR-mediated compensation of the inhibitory PP2C expression (Tischer *et al*, 2017), or purified proteins were analyzed by the ABA receptor-phosphatase assay (Fuchs *et al*, 2014). The yeast analysis revealed a gradual activation of the ABA response with increasing external ABA concentration for all RCAR–PP2C combinations. However, there were clear differences between RCAR12 and RCAR13 such that RCAR13 required lower ABA levels to achieve half-maximum activation (Fig 3A–C). The differences among the three PP2Cs were minor. In *Arabidopsis* protoplasts, ectopic expression of RCAR12 and RCAR13 stimulated ABA signaling in response to exogenous ABA concentration and again RCAR13 activated the ABA response more sensitively than RCAR12 (Fig 3D–F). The *in vitro* analysis corroborated the higher ABA sensitivity of RCAR13 compared to RCAR12 as observed in yeast and protoplasts (Fig 3G–I). The effective ABA levels in the three assay systems differed. Half-maximum regulation in yeast was observed in the μM ABA range, while in protoplasts, submicromolar concentrations of ABA were sufficient, and even lower ABA levels *in vitro*. The high external ABA levels required in the yeast system might be explained by an inefficient ABA uptake resulting in low endogenous concentrations. Analysis for ABA in yeast indicated the absence of the ligand without exogenous ABA provision, and exposure of yeast cells to 10 μM ABA resulted in an intracellular ABA level of 0.13 μM (Fig EV3), explaining the requirement of high exogenous ABA concentrations for activation of the response pathway. The comparative analyses in yeast, protoplasts, and *in vitro* consistently revealed higher ABA sensitivity of RCAR13 compared with RCAR12. The results indicate that despite the major differences of the assay

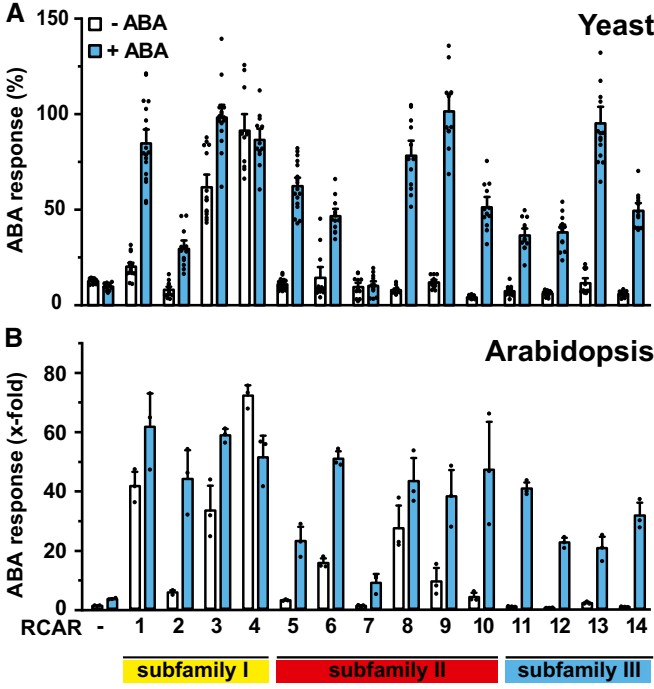

**Figure 2. Activation of ABA signaling by the *Arabidopsis* receptor family members in yeast and plant protoplasts.**

A  In yeast, the OST1- and ABF2-mediated reporter activity was inhibited by ABI1 (−RCAR). The subsequent recovery of the response was determined by co-expressed ABA receptors RCAR1-RCAR14 belonging to three subfamilies, in the presence or absence of 30 μM ABA. The ABA response is expressed relative to the response obtained by replacing ABI1 with the catalytically inactive ABI1^D177A.

B  Regulation of an ABA-responsive LUC reporter in *Arabidopsis* protoplasts by ectopic expression of 1 μg ABI1 effector in combination with 5 μg ABA receptor DNA (±10 μM ABA) per sample containing $10^5$ protoplasts.

Data information: In (A), each bar represents the mean ± SEM; $n = 12$, biological replicates derived from four independent yeast transformants; for RCAR1/RCAR5 and RCAR3/RCAR13, 18 and 15 biological replicates were used, respectively. Outliers detected by Grubbs test were removed from the analysis. In (B), each bar represents the mean ± SEM of three independent transformations normalized to the activity of a control samples without RCAR and ABA.

Source data are available online for this figure.

systems, the analysis of ABA signaling in yeast allows recapitulating properties of ABA receptor complexes and their effects on the ABA signal pathway.

**Regulation of ABA-responsive transcription factors by SnRK2s**

The previous analysis of OST1-mediated ABF transactivation (Fig 1A) showed a clear preference for ABF4 and major differences in the efficiency of ABF transactivation. The finding could imply that SnRK2 members differently target ABFs. To test this hypothesis, we expressed all 10 SnRK2s belonging to three different subgroups (Fig 4A) together with ABF2 in yeast. OST1 and the four subgroup I members SnRK2.1, SnRK2.4, SnRK2.5, and SnRK2.10 induced reporter expression (Fig 4B). Transactivation of ABF2 by the subgroup I members was more efficient than by OST1. No transactivation was observed for subgroup II, and other subgroup III

members though immunodetection indicated correct SnRK2 expression (Fig 4B, inset). Analysis of ABF1 to ABF4 showed different efficiencies in their activation by SnRK2.1, SnRK2.4, SnRK2.5, and SnRK2.10 but similar preferences among the active protein kinases (Fig 4C). SnRK2.2 and SnRK2.3 did not transactivate any ABF. Normalization of the reporter activities by the activation capacity of the ABFs fused to the yeast GAL4 activation domain without SnRK2 expression (Fig EV1B) confirmed the ABF2- and ABF4-preferred transactivation by the active SnRK2s in yeast (Fig EV4).

The architecture of SnRK2s includes regulatory domains such as the ABA-box, also called DII, and an osmoregulatory domain DI that are schematically shown for OST1 (Fig 4D; Belin *et al*, 2006; Yunta *et al*, 2011). Deletion of the ABA-box affects OST1 activity and ABA-dependent regulation (Yoshida *et al*, 2006). In yeast, partial deletion of the ABA-box in OST1 increased the transactivation efficiency and complete removal of this regulatory domain (Δ320) stimulated the ABA response sixfold (Fig 4E). Deletion of the ABA-box in other SnRK2 members resulted in a moderate increase in reporter expression in subgroup I members, while SnRK2.2 and SnRK2.3 remained inactive (Appendix Fig S2A). Removal of the ABA-box did not affect the efficiency of ABI1 to inhibit the OST1-mediated transactivation (Appendix Fig S2B). Phosphorylation of the activation loop of OST1 at serine 175 or threonine 176, and threonine 179 (Fig 4D) is known to be required for SnRK2 activity (Vlad *et al*, 2010; Ng *et al*, 2011). The inactivity of SnRK2.2 and SnRK2.3 in yeast might be the consequence of inefficient auto- or transactivation of the activation loop. To examine this possibility, OST1, SnRK2.3, and SnRK2.4 were analyzed for posttranslational modification in yeast by targeted phosphosite analysis (Figs 4F and EV5, and Appendix Fig S3). The activation loop is highly conserved among the three SnRK2s (Fig EV5A), and phosphorylation of the corresponding sites was observed for OST1, and at higher levels for SnRK2.4 but essentially not for SnRK2.3 (Fig 4F) consistent with the SnRK2 activities in yeast. Dephosphorylation of serine 175 of OST1 entails the inhibitory action of ABI1 (Soon *et al*, 2012; Xie *et al*, 2012). Comparative analysis of OST1 and SnRK2.4 for inhibition by ABA coreceptors showed that all PP2Cs reduced SnRK2-mediated reporter activation. SnRK2 inhibition was efficient with ABI1-related PP2Cs, less though with PP2CA-related enzymes especially AHG1, and SnRK2.4 was less sensitive to the control by PP2CA-related coreceptors compared to OST1 (Fig 4G). Analysis of all active SnRK2s revealed overall a similar efficiency of ABI1 to inhibit the ABF transactivation (Fig EV2). Taken together, several subgroup I SnRK2s appear to be regulated by ABA coreceptors, and they preferentially activate ABF2 and ABF4 over ABF1 and ABF3 similar to OST1 in yeast.

**Transactivation by SnRK2s differs in their ABA-sensitive regulation**

The impact of different SnRK2s on the interaction of the receptor components RCAR and PP2C is not known. Based on structural and biochemical analyses, OST1 interacts via its catalytic domain and the ABA-box with the PP2C HAB1 (Soon *et al*, 2012). Both the protein kinase and the ABA receptor compete for the catalytic cleft of the PP2C, and the interaction of RCAR and PP2C is stabilized by ABA. Hence, different SnRK2s might differ in their ability to compete with RCARs, which is modulated by ABA and possibly

    

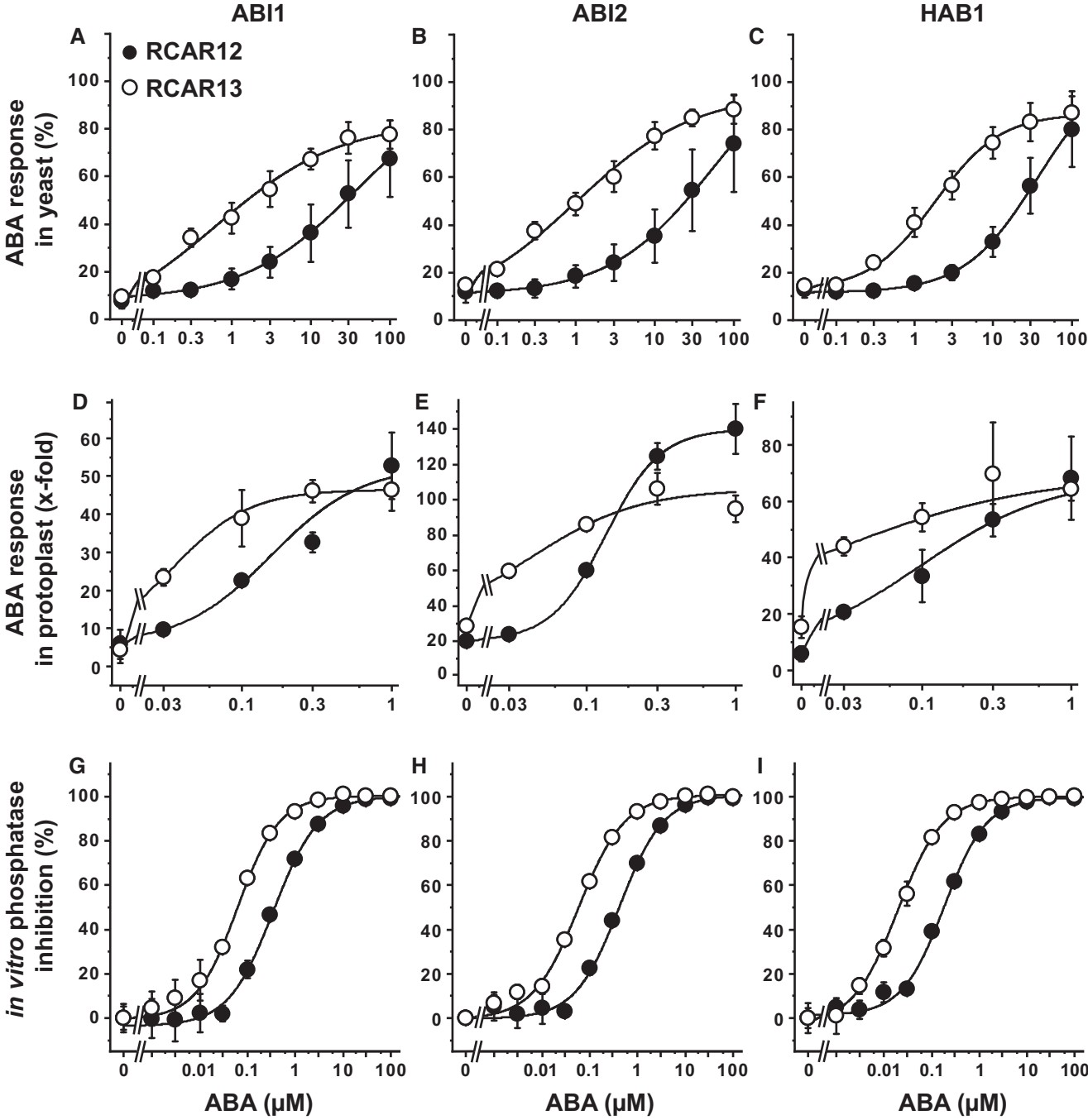

**Figure 3. Comparison of ABA coreceptor regulation by RCAR12 and RCAR13 in yeast, *Arabidopsis* protoplasts, and *in vitro*.**

The ABA-dependent regulation of the PP2Cs ABI1 (A, D, G), ABI2 (B, E, H), and HAB1 (C, F, I) by RCAR12 and RCAR13 was analyzed in yeast (A–C), *Arabidopsis* protoplasts (D–F), and *in vitro* (G–I).

A–C  The analysis in yeast was conducted as mentioned in Fig 2A except that other PP2C–RCAR combinations were used and the ABA concentrations varied as indicated.

D–F  Analysis of protoplasts was performed in the ABA-deficient background *aba2-1* in the presence of different exogenous ABA concentrations. The ABA response was approximately tenfold inhibited by ectopic expression of the respective PP2Cs (Tischer *et al*, 2017). The induction of the reporter gene is related to protoplasts without ectopic RCAR expression and in the absence of exogenous ABA.

G–I  The *in vitro* analysis was performed by using 50 nM PP2C and 100 nM RCAR at different ABA concentrations. Maximum inhibition of PP2C activity by ABA was set to 100%.

Data information: In (A–C), each data point represents the mean ± SD; *n* = 9 biological replicates derived from three independent yeast transformants. In (D–F), data represent the mean ± SD of three independent transfections. In (G–I), data represent the mean ± SD of three replicates.
Source data are available online for this figure.

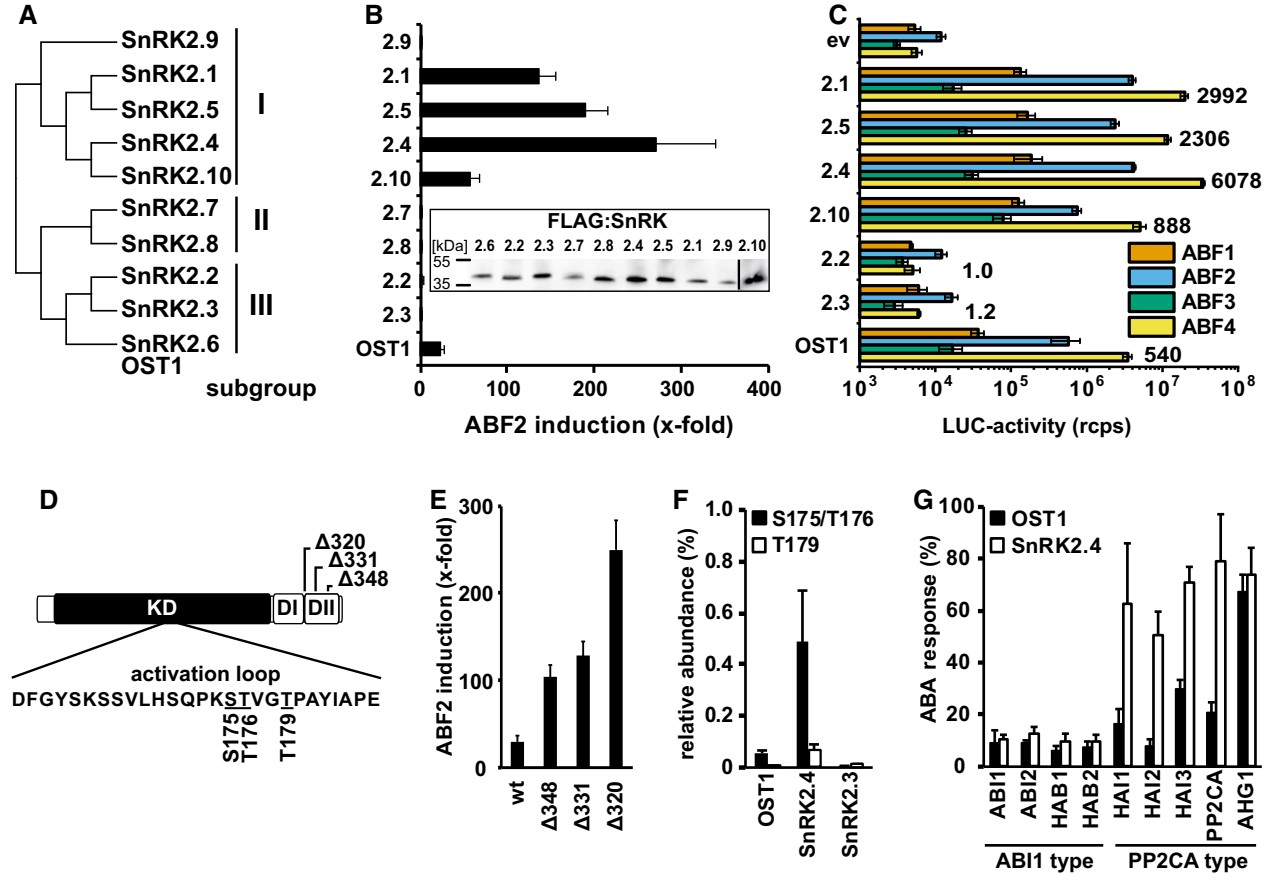

**Figure 4. Regulation of the ABA response by different protein kinases of the SnRK2 family in yeast.**

A Dendrogram of the *Arabidopsis* SnRK2 protein kinases belonging to three subgroups.

B The expression of the SnRK2s in yeast revealed ABF2-mediated transactivation by SnRK2.1, SnRK2.4, SnRK2.5, and SnRK2.10 in addition to OST1. Inset: Immunodetection of FLAG-tagged SnRK2s expressed in yeast.

C The active subgroup I members together with SnRK2.2 and SnRK2.3 were analyzed for differential activation of ABFs. The OST1 data shown in Fig 1A are included for better comparison. The induction level of ABF4 relative to the empty vector control (ev) is indicated.

D The OST1 domain structure comprises the kinase domain (KD) containing the activation loop and the two carboxyterminal regulatory domains DI and DII/ABA-box. Potential phosphorylation sites within the activation loop are underlined.

E Increased ABF2-mediated reporter expression by OST1 harboring an ABA-box deletion as indicated in (D).

F Analysis of the phosphorylation status in the activation loop of SnRK2s expressed in yeast. Phosphorylation occurred at serine 175 or threonine 176, and at threonine 179. The relative abundance is given as the ratio of signal intensity of phosphorylated versus non-phosphorylated peptide for OST1, SnRK2.3, and SnRK2.4 in the mass spectrometric analysis. The numbering of phosphosites corresponds to OST1. The peptide sequences of the SnRK2 activation loop and the details for phosphosite detection and quantification are provided in Appendix Fig S3 and Fig EV5.

G Inhibition of OST1 and SnRK2.4 by clade A PP2Cs.

Data information: The data in (B, C, E, G) represent the mean ± SEM; *n* = 6 biological replicates derived from three independent yeast transformants. The data in (B) and (E) are normalized to control samples expressing ABF2 but no SnRK2 and (G) to cells expressing inactive ABI1$^{D177A}$ instead of ABI1. Data in (F) represent the mean ± SD of four biological replicates.

Source data are available online for this figure.

affect ABA-response sensitivity. To address this question, different SnRK2s were analyzed in combination with RCAR11 and ABI1 at different ABA levels in yeast. Indeed, the ABA sensitivity of reporter regulation varied largely depending on the expressed SnRK2 member (Fig 5A and B). The response recovery by ABA was in most cases partial (Fig 5C). The highest exogenous ABA concentration (0.1 mM) fully restored ABA signaling with SnRK2.5 while the recovery for other SnRK2s varied from approximately 47% for OST1 to about 86% for SnRK2.10.

The deduced ABA values for apparent half-maximum recovery varied considerably. They were lowest for SnRK2.1, SnRK2.5, and

SnRK2.10 and ranged between 0.3 and 0.6 μM exogenous ABA (Fig 5A and B). The values were approximately 5 μM ABA for SnRK2.4 and 100 μM ABA for OST1. Hence, OST1 required approximately 20 times higher ABA levels for half-maximum response recovery and was at that point more than 20 times less efficient in transactivating ABF2 than SnRK2.4. For a more detailed assessment, we focused on OST1 and SnRK2.4 and analyzed the ABA-dependent response in different ABA receptor-ABI1 interactions (Fig 5D–F). The analysis with RCAR1, RCAR8, and RCAR14 showed a consistently higher ABA sensitivity of the response in SnRK2.4- versus OST1-expressing yeasts at comparable expression levels

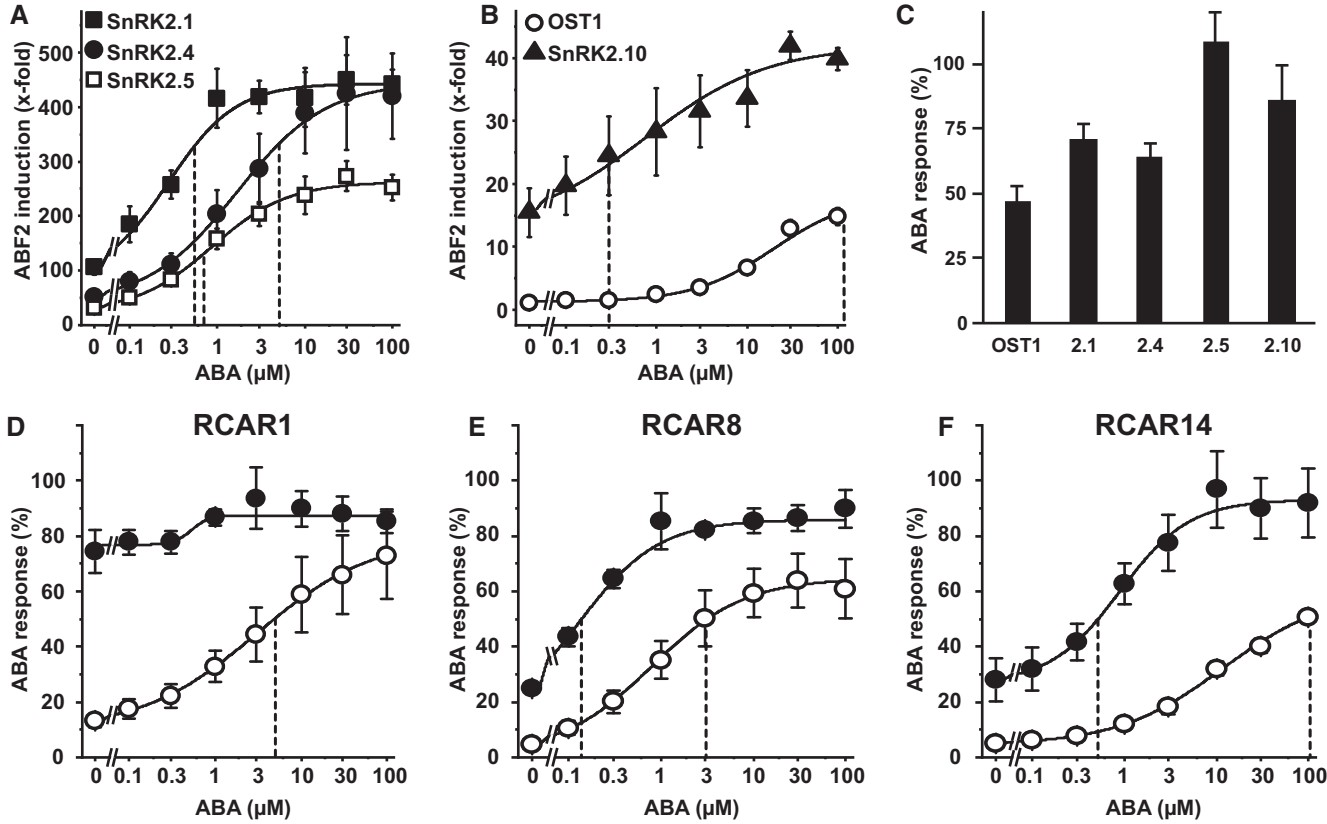

**Figure 5. Different ABA sensitivity conferred by SnRK2 members in yeast.**

A, B ABF2-mediated reporter induction by different SnRK2s was assessed in the presence of the receptor complex RCAR11 and ABI1, and varying exogenous ABA concentrations.

C Maximum ABA-response recovery of samples shown in (A, B) relative to the reporter activity of 0.1 mM ABA-treated yeast cells expressing SnRK2s but no ABI1.

D–F Analysis of the ABA response mediated by OST1 and SnRK2.4 using the receptors RCAR1 (D), RCAR8 (E), and RCAR14 (F).

Data information: The ABA concentration that provides half-maximum response recovery is indicated by dashed vertical lines. The data in (A–C) represent the mean ± SD; $n = 6$ biological replicates derived from three independent yeast transformants. Replicates were normalized to controls without SnRK2 in (A, B) and to controls without ABI1 and RCAR for each SnRK2 in (C). The data in (D–F) are the mean ± SEM; $n = 12$, biological replicates from four independent yeast transformants for OST1 + RCAR1/RCAR8, and SnRK2.4 + RCAR8. The other combinations were analyzed in nine biological replicates derived from three independent yeast transformants.

Source data are available online for this figure.

(Appendix Fig S4). Interestingly, RCAR1 stimulated the SnRK2.4-mediated ABA response already without ABA to more than 70%, while the OST1-mediated ABA response required 0.1 mM exogenous ABA to achieve a similar relative ABF2 activation (Fig 5D). The same trend was evident for RCAR8 and RCAR14. The higher ABA sensitivity of the SnRK2.4-driven ABA signaling could be explained by a more efficient displacement of the protein kinase from the inhibitory PP2C by the ABA receptor in comparison with OST1.

**SnRK2 interaction with ABI1 controlled by ABA receptors and ABA**

The effect of ABA receptors on the protein interaction between SnRK2 and ABI1 was analyzed by fluorescence resonance energy transfer in combination with fluorescence lifetime imaging (FRET/FLIM). SnRK2.4 and OST1 were expressed as GFP fusion proteins in leaves of *Nicotiana benthamiana*, and the GFP fluorophore revealed

similar fluorescence lifetimes for both proteins *in situ* (Fig 6A). In the presence of co-expressed *Arabidopsis* ABI1 that was tagged by mCherry fluorophore, the GFP lifetime of both SnRK2s was reduced by about 0.2 ns supporting similar interaction efficiencies of both protein kinases (Long *et al*, 2017; Weidtkamp-Peters & Stahl, 2017). Additional expression of *Arabidopsis* RCAR1 affected the GFP lifetime of both protein kinases differently. In the absence of exogenous ABA, a significant increase in the GFP lifetime for GFP:SnRK2.4 was observed that indicated the disturbance of the ABI1-SnRK2 interaction, but not for GFP:OST1 (Fig 6B). Foliar administration of ABA (0.1 mM) resulted in an increase in GFP lifetimes for OST1 similar to the level observed for SnRK2.4 without exogenous ABA (Fig 6A). The results support a more efficient displacement of SnRK2.4 from ABI1 by RCAR1, i.e., a more stable interaction of ABI1 with OST1 compared to SnRK2.4. RCAR1 is a high-affinity ABA receptor of subfamily I and replacing it with RCAR14 belonging to the lower affinity subgroup III corroborated the different interaction strengths

of OST1 and SnRK2.4 with ABI1. In the presence of RCAR14, both protein kinases required exogenous ABA for efficient displacement from ABI1; however, clearly higher ABA levels were necessary for OST1 (Fig 6C and D).

## Discussion

Phytohormone signals integrate environmental cues into plant metabolism and development. Two prominent phytohormones, auxin and ABA, are unique in being perceived by a large number of

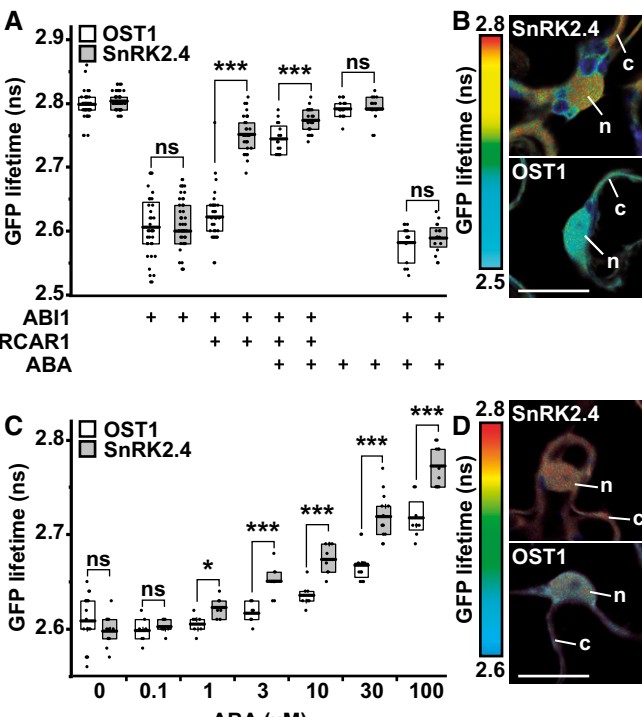

**Figure 6. Comparison of OST1 and SnRK2.4 in the interaction with ABI1 affected by ABA receptors and ABA.**

A   FRET-FLIM analysis of GFP:OST1 or GFP:SnRK2.4 and the effect of ABI1 fused to the mCherry fluorophore in the presence of RCAR1 and ABA. GFP lifetimes were determined after transient expression of the *Arabidopsis* proteins in leaves of *Nicotiana benthamiana*.

B   Representative false color images of GFP:SnRK2 lifetimes measured in leaf tissues expressing OST1 or SnRK2.4 in combination with mCherry:ABI1 and RCAR1 without exogenous ABA. The bar marks 20 μm and the nucleus (n) and cytosol (c) are indicated.

C   Analysis of SnRK2-ABI1 interaction as mentioned in (A) but in the presence of RCAR14 and different exogenous ABA levels. RCAR14 requires ABA for efficient displacement of SnRK2s from ABI1.

D   Representative GFP lifetime false color image of the analysis shown in (C) 2 h after 30 μM exogenous ABA administration. Details are as described in (B).

Data information: The single data points shown in (A, C) were collected from three independent experiments. Details about the analyzed *n*- and *P*-values are provided in the source data file. Boxes show the quartiles and the mean. Stars indicate significance of two-tailed Student's *t*-test with $P < 0.05$ (*), $< 0.001$ (***).

Source data are available online for this figure.

different receptor complexes (Parry *et al*, 2009; Pierre-Jerome *et al*, 2014; Tischer *et al*, 2017). In the ABA-response pathway, the complexity is further increased by protein kinases that function as mediators between receptor complexes and cellular targets such as transcription factors. As a consequence, deciphering the contribution of signaling components to response specificity and sensitivity is a major challenge.

In this study, we analyzed the complexity of ABA signaling using a reductionist approach. The multitude of compatible components encompassing different ABA receptors, coreceptors, protein kinases, and ABA-responsive transcription factors was systematically analyzed for functionality and ABA responsiveness in yeast. This was achieved by comparing ABA core signaling pathways built from single components for each signaling step. A modular cloning system (based on Binder *et al*, 2014) was used for expressing wild-type proteins and generating the combinatorics of ABA signaling pathways as a systems biology tool.

The analysis in yeast enabled us to address the ABA responsiveness conferred by different ABA receptors in a heterologous host. We were able to assess whether SnRK2 protein kinases other than subfamily III function in ABA signaling and affect the sensitivity of the ABA signal relay, and whether SnRK2s selectively activate ABF transcription factors. Comparative analysis of all 14 ABA receptors in yeast and *Arabidopsis* protoplasts corroborated the different ABA sensitivities conferred by receptor subfamilies as observed in plant cells (Tischer *et al*, 2017). The absence of ABA in yeast (Fig EV3) allowed us to determine whether ABA receptors can activate ABA signaling without this ligand. Indeed, the subfamily I members RCAR1, RCAR3, and RCAR4 activated ABA signaling in the absence of exogenous ABA. The results are in agreement with the analysis of the ABA receptors in ABA-deficient plant protoplasts having strongly reduced ABA levels (Tischer *et al*, 2017). In contrast, ABA receptors of subfamily III required ABA in yeast and plant protoplasts (Figs 2 and 3). The high consistency of the results from fungal and plant analyses is surprising given the fact that only a single PP2C, OST1, and ABF2 were expressed in yeast, whereas several members of the signaling components are anticipated to be present in a single-plant cell as exemplified for guard cells (Wang *et al*, 2011). An explanation for the observed consistency might be the similar regulation of and by SnRK2s acting as central mediators of the response pathway. This view is supported by the comparable inhibition of SnRK2s via the ABA coreceptor ABI1 (Fig EV2) and by the conserved SnRK2 preferences for ABF activation with the order of highest preference for ABF4, followed by ABF2, ABF3, and ABF1 (Fig 4C).

OST1 has an autoactivating capacity, while the other subgroup III members SnRK2.2 and SnRK2.3 require transactivation (Boudsocq *et al*, 2007; Ng *et al*, 2011; Cai *et al*, 2014). Consistently, SnRK2.2 and SnRK2.3 were not active in yeast and possibly require the additional expression of an upstream-acting protein kinase in yeast. Candidates for such an activator might be the protein kinases BIN2 (Cai *et al*, 2014) and a mitogen-activated protein kinase kinase kinase shown to activate SnRK2s under hyperosmotic stress in the moss *Physcomitrella patens* (Saruhashi *et al*, 2015; Amagai *et al*, 2018). Besides OST1, the subgroup I members SnRK2.1, SnRK2.4, SnRK2.5, and SnRK2.10 stimulated ABA signaling in our analysis. Subgroup I SnRK2s are activated under salt and hyperosmotic stress and emerged as important regulators of root growth including fine-

tuning of plant growth by posttranslational mRNA decay of drought-induced genes (Fujii *et al*, 2011; McLoughlin *et al*, 2012; Soma *et al*, 2017).

Our yeast experiments suggest that subgroup I SnRK2s have the capacity for auto-activation similar to OST1. OST1 activation entails phosphorylation of serine 175 located within the activation loop (Xie *et al*, 2012). The activation loop is conserved among SnRK2s, and analysis in yeast showed phosphorylation of this serine residue in OST1 and a several fold higher phosphorylation status in SnRK2.4 but a low phosphorylation level in SnRK2.3, which is in contrast to the phosphorylation level of the SnRK2.3 in plant tissues (Kline *et al*, 2010). The phosphorylation status of the SnRK2 in the activation loops found in yeast clearly correlated with their transactivation capacity. OST1 revealed moderate activity compared to the active subgroup I SnRK2s. Responsible for the attenuated OST1 activity was a carboxyterminal domain called ABA-box that stabilizes the interaction of SnRK2 and PP2C. Deletion of this domain increased transactivation activity of OST1 to comparable levels as for subgroup I members with or without the ABA-box (Fig 4E, Appendix Fig S2A). The activation occurred independent of the inhibitory PP2C, indicating that this domain constrains OST1 auto-activation. The results are consistent with the observation in *Arabidopsis* of higher response activation by OST1 with non-functional ABA-box (Yoshida *et al*, 2006). The release of SnRK2s from attenuation is also modulated by ABA receptors as indicated by the hyper-activation of SnRK2s in a multiple RCAR-deficient *Arabidopsis* mutant under osmotic stress (Zhao *et al*, 2018).

In our study, the ABA sensitivity of the signal relay was affected by the receptor complex and by the SnRK2 member. The subgroup I SnRK2s were more sensitively activated by ABA in the presence of RCAR11-ABI1 than OST1, and the half-maximum ABA-mediated activation was shifted by a factor of 20 for SnRK2.4 and more than 100 for SnRK2.1, SnRK2.5, and SnRK2.10 to lower ABA levels compared to OST1. Structural and biochemical analyses revealed a competing interaction between OST1 and ABA receptor for binding to the catalytic cleft of the PP2C as substrate and pseudosubstrate, respectively (Soon *et al*, 2012; Xie *et al*, 2012). Our comparative analyses with OST1 and SnRK2.4 are in agreement with this concept and showed that they differ with respect to the efficiency with which they compete with ABA receptors. SnRK2.4 was more easily released from ABI1 and more potent in transactivating ABF2 than OST1, which resulted in a higher ABA responsiveness conferred by SnRK2.4. The ABA receptors RCAR1, RCAR8, RCAR11, and RCAR14 released SnRK2.4 at lower ABA levels than OST1 from ABI1 inhibition in yeast. Consistently, protein interaction analysis in tobacco leaves supported a more efficient displacement of SnRK2.4 from ABI1 compared with OST1 in the presence of RCAR1 and RCAR14. Even in the absence of ABA, the RCAR1 receptor was able to release SnRK2.4 from PP2C inhibition, which can be explained by the high affinity of RCAR1 to bind to ABI1 (Ma *et al*, 2009; Tischer *et al*, 2017). Hence, the molecular basis for the difference observed between OST1 and SnRK2.4 appears to be a more facile release of SnRK2.4 from ABI1 inhibition in the presence of ABA receptors and ABA allowing the transactivation of downstream ABFs. There were differences between OST1 and SnRK2.4 in the degree of inhibition by clade A PP2Cs. SnRK2.4 and OST1 were comparably inhibited by the ABI1-related subclass of ABA coreceptors including ABI2, HAB1, and HAB2 but SnRK2.4 was less efficiently regulated by PP2CA-type

phosphatases encompassing PP2CA, HAI1 to HAI3, and AHG1. The finding is consistent with the reported inhibition of SnRK2.4 by ABI1 and PP2CA (Krzywińska *et al*, 2016b).

In summary, the comparative analysis of ABA signaling modules in yeast revealed that the subgroup I members SnRK2.1, SnRK2.4, SnRK2.5, and SnRK2.10 known to be activated under hyperosmotic stress have the hallmarks for being integral parts of the ABA signal pathway. Comparable to the prototype OST1, they were inhibited by ABA coreceptors, ABA-dependently regulated by ABA receptors, and they activated ABA-responsive transcription factors.

The reconstruction of the transcriptional signaling pathway of ABA in yeast provides a valuable tool for future studies on activating signals and the molecular function of its components. ABA and other small molecules including xenobiotics are taken up into yeast (Park *et al*, 2009; Krajewski *et al*, 2013), and hence, the system allows one to screen for ABA receptor ligands and to characterize their receptor specificity (Okamoto *et al*, 2013; Takeuchi *et al*, 2014). Distinct ABA receptors emerged as suitable targets for improving water use in plants (Yang *et al*, 2016; Mega *et al*, 2019) and identifying ABA agonists that selectively target such ABA receptors holds promise to improve the efficiency of water use in crops for a more sustainable agriculture (Blankenagel *et al*, 2018).

# Materials and Methods

## Materials

Chemicals were obtained from Sigma-Aldrich (www.sigmaaldrich.com) and J. T. Baker (www.fishersci.de), and (S)-ABA from CHEMOS (www.chemos-group.com).

## Construction of yeast ABRE reporter strain

The functional ABRE TACGTGGC and the non-functional ABRE TTTAAGGC were fused as two tandem repeats upstream of the 35S minimal promoter (−45 b) driving the expression of the *firefly* luciferase (LUC; Fig EV1A, Appendix Table S1). The reporter gene comprising the 4xABRE:-45bp35S::LUC::tNOS reporter was ligated via the NotI and SalI sites into pIS385-LYS2 (Sadowski *et al*, 2007) and integrated into the *LYS2* locus of *Saccharomyces cerevisiae* strain BMA64-1A (EUROSCARF, Uni-Frankfurt, Germany). Integration and URA3 marker rescue by targeted locus disintegration (Sadowski *et al*, 2007) were confirmed by growth on plates supplemented with 5-fluoroorotic acid (1 mg/ml). Successful generation of the yeast ABRE reporter strain was confirmed by lysine auxotrophy and DNA sequence analysis of the *LYS2* locus.

## Yeast constructs and transformation

The GAL1 promoter (pGAL1) of pGREG503 (Jansen *et al*, 2005) was replaced by pADH1 from pGBT9 (Clontech; GenBank Accession #U07646) and was used for the expression of ABF1 (AT1G49720), ABF2 (AT1G45249), ABF3 (AT4G34000), ABF4 (AT3G19290), and ABI5 (AT2G36270). For aminoterminal AD fusions, ABF1–ABF4 and ABI5 were cloned into pGAD424 (Clontech, GenBank Accession #U07647). Endogenous BpiI and BsaI sites present in cDNAs of SnRK2s (Boudsocq *et al*, 2004), clade A PP2Cs, or RCARs (Tischer

*et al*, 2017) were removed by synonymous site-directed mutagenesis (Fuchs *et al*, 2014), and the obtained cDNAs were cloned into level I (LI_BpiI) vector as described (Binder *et al*, 2014). ABA-box truncated SnRK2 genes and CPK4 with deleted calmodulin-like domain (Zhu *et al*, 2007; Wernimont *et al*, 2010) were generated from LI vectors by deletion at amino acid positions 348, 331, and 320 for OST1 and for other SnRK2s (SnRK2.1$^{\Delta304}$, SnRK2.2$^{\Delta322}$, SnRK2.3$^{\Delta321}$, SnRK2.4$^{\Delta331}$, SnRK2.5$^{\Delta305}$, SnRK2.7$^{\Delta308}$, SnRK2.8$^{\Delta299}$, SnRK2.9$^{\Delta307}$, SnRK2.10$^{\Delta305}$) and 306 for CPK4 (CPK4$^{\Delta306}$). As a control for ABF activation by SnRK2s, the mutant OST1$^{S175A}$ was used which is impaired in auto-activation (Belin *et al*, 2006). All LI vectors were verified for correctness by DNA sequence analysis. SnRK2 expression constructs were assembled in the level II vector (LII) pYEAST_LII_3-4_CEN_TRP with GAL1 promoter and TDH1 terminator (tTDH1) with or without an aminoterminal 3xFLAG-tag. Empty cloning positions were filled with dummy elements (Binder *et al*, 2014). If not stated otherwise pYEAST_LII_1-2_CEN_LEU vector with GK1 promoter (pGK1) and TDH1 terminator, or pYEAST_LII_5-6_CEN_URA vector with pTDH3 and tTDH1 was used for the expression of PP2Cs and RCARs, respectively. Yeast transformation and colony selection were performed according to Amberg *et al* (2005).

## Yeast luciferase assay

The LUC activity was assayed in life yeast cells in the presence of luciferin (Leskinen *et al*, 2003). Briefly, independently freshly transformed yeast colonies were inoculated in 0.5 ml selective minimal SD medium containing 2% glucose supplemented with the auxotrophic amino acids and were cultivated overnight in a gyratory shaker (200 rpm, 30°C, Thermoshake-THO500, Gerhardt, Königswinter, Germany). After addition of 4.5 ml SD medium, the culture was allowed to reach OD$_{600}$ between 0.6 and 0.8, cells were sedimented (1,500 *g*, 5 min), and the cell density was adjusted to OD$_{600}$ of 10. For ABA titration experiments, the culture was upscaled to 15 ml culture volume. Induction of SnRK2 expression under the control of the GAL1 promoter was achieved by inoculation of 250 μl SD medium containing 2% galactose to an OD$_{600}$ of 0.2 and subsequent cultivation for 16–18 h in the presence or absence of ABA. Yeasts were grown in flat bottom 96-well plates (CytoOne; Starlab, Hamburg, Germany) at 30°C and 200 rpm. Single colony-derived samples that showed no growth on induction plates were omitted from subsequent analysis. LUC activity and cell density of 100 μl cell suspension (about $3 \times 10^5$ cells) were determined. Light emission of yeasts was measured in a HIDEX plate luminometer (Turku, Finland) after injection of 100 μl freshly prepared luciferin substrate (1 mM D-luciferin potassium salt; Promega) provided in 100 mM citric acid, pH 3 (Leskinen *et al*, 2003) for 5s and was corrected for background. The LUC activity was normalized to the cell density (Abs$_{600}$) measured at 600 nm in 96-well flat plates (Sarstedt, Nümbrecht, Germany). LUC activity is given as the ratio of light emission and Abs$_{600}$ in relative counts per second (rcps).

## Transient ABA response in *Arabidopsis* protoplast

Preparation and analysis of *Arabidopsis* protoplasts and effector DNAs were performed as described previously (Tischer *et al*, 2017).

For RCAR-mediated activation of ABA signaling, *Arabidopsis* protoplasts from Columbia wild-type accession (Col-0) were used. For ABA titration experiments, protoplasts of the ABA-deficient *aba2-1 Arabidopsis* mutant were used. Briefly, $10^5$ protoplasts were transfected with 5 μg pRD29B::LUC reporter construct, 3 μg p35S::GUS control reporter for internal expression normalization, 5 μg p35S::RCAR effector DNA and indicated amounts of p35S::PP2C effector DNA. 40, 60, and 50 ng of ABI1, ABI2, and HAB1 effector DNA were used to inhibit the basal ABA response to approximately 10%, respectively. LUC activity was determined after incubation at 25°C for 16–18 h in the presence of indicated exogenous ABA concentrations.

## *In vitro* receptor assay

Expression, purification, and analysis of RCAR and PP2Cs proteins were as described (Kepka *et al*, 2011). pET-24a(+)-RCAR12, pET-24a(+)-RCAR13, pQE70-HAB1, pSEVA431_LII_6xHIS:MBP:ABI1, and pSEVA431_LII_6xHIS:MBP:ABI2 were transformed in *Escherichia coli* strain Rosetta(DE3)pLysS, and proteins were expressed and purified to homogeneity. In brief, *E. coli* were grown at 37°C in terrific broth media to an optical density at 600 nm of approximately 1. Subsequently, protein expression was induced with administration of 1 mM isopropyl-β-D-thiogalactoside for 4 h (RCAR proteins) and 2.5 h (PP2C proteins). Cells were lysed by sonication in 100 mM Tris–HCl, 100 mM NaCl, 1 mM DTT, 20 mM imidazole, pH 7.9, and proteins were purified with HisTrap FF Crude protein purification columns (GE Healthcare, Chicago, USA). Elution was performed with 100 mM Tris–HCl buffer containing 100 mM NaCl, 1 mM DTT, 10% glycerol, pH 7.9, supplemented with 250 mM imidazole. The buffer without imidazole was subsequently used for dialysis of the protein solutions. The receptor assay was performed with 50 nM PP2C and a twofold molar excess of RCAR. In this assay, the phosphatase activity of ABI1, ABI2, and HAB1 was reduced by RCAR12 in the presence of saturating ABA levels (0.6 mM) to a residual activity of 17, 14, and 8%, respectively, and the inhibitions were set to 100%. The phosphatase activity was measured using 4-methyl-umbelliferyl-phosphate as a substrate (Ma *et al*, 2009). The increase in fluorescence was recorded over 15 min with excitation and emission wavelengths at 360 and 460 nm, respectively, using a Synergy 2 plate reader (Bio-Tec, Winooski, USA).

## Immunodetection of SnRK2 proteins

Yeast cells (5 ml culture) expressing *Arabidopsis* SnRK2 proteins that were aminoterminally tagged with three FLAG epitopes (3xFLAG:SnRK2) were washed in 1 ml of 10 mM EDTA solution, pH 8, and cells were resuspended in 100 μl yeast lysis buffer (YLB; 50 mM Tris–HCl, pH 8; 1% DMSO; 100 mM NaCl, 1 mM EDTA). Cells were disrupted with 200-μl glass beads (0.2–0.4 mm, Merck, Darmstadt, Germany) in a swing mill (30 Hz, 4 × 1 min, 4°C, Tissue Lyser II, Qiagen, Hilden, Germany). After removal of cell debris by centrifugation (20,000 *g*, 20 min, 4°C), the supernatant was diluted to 1 μg/μl total protein and aliquots were used for dot blot or Western blot analysis. For dot blot analysis, samples were diluted twofold with 0.2 N NaOH solution, spotted on nitrocellulose membrane (pore size 0.45 μm; Sartorius, Göttingen, Germany), and dried at 60°C. For Western blot analysis, proteins were separated by

electrophoresis in a 12% SDS-polyacrylamide gel and transferred to the nitrocellulose membrane by semidry blotting. The FLAG-tagged proteins were detected with monoclonal ANTI-FLAG® M2 antibody (Sigma-Aldrich, #F1804) diluted 1:4,000 and goat anti-mouse IgG-HRP conjugate (Immunoreagents, Raleigh, USA) diluted 1:1,000 in buffer containing 25 mM Tris–HCl, 150 mM NaCl, 0.3% Tween 20, pH 7.4, with 5% milk powder (TSI, Zeven, Germany).

## Confocal microscopy

Leaves of 5-week-old *N. benthamiana* were infiltrated with an equal mixture of *Agrobacterium tumefaciens* GV3101 (MP90) expressing the viral protein p19 and the different *Arabidopsis* proteins on a single T-DNA (Sparkes *et al*, 2006). Plants were grown under long day condition (16-h light, 180 μE/m²/s), and the abaxial side of leaves was inoculated with the *Agrobacterium* culture using a 2-ml syringe. The bacteria contained binary level III plasmids LIII_A-B (Binder *et al*, 2014) and allowed the expression of GFP:OST1, GFP:SnRK2.4, mCherrry:ABI1, RCAR1, and RCAR14, and combinations thereof, under the control of the viral 35S promoter. Infiltrated plants were incubated for 3 days and sprayed with indicated concentrations of ABA in 0.1% DMSO, 0.01% Tween 20, 2 h prior confocal microscopy. For confocal microscopy and FRET-FLIM analysis, images were taken using an Olympus FluoView™ 3000 inverse laser scanning confocal microscope with an UPLSAPO 60XW 60×/NA 1.2/WD 0.28 water immersion objective (Olympus, Hamburg, Germany). For imaging of the GFP and mCherry fluorophore, tissue samples were excited at 488 and 561 nm, respectively. For FRET-FLIM data acquisition, the PicoQuant advanced FCS/FRET-FLIM/rapidFLIM upgrade kit (PicoQuant, Berlin, Germany) was used. GFP was excited at 485 nm with a pulsed laser (pulse rate 40 mHz, laser driver: PDL 828 SEPIA II, laser: LDH-D-C-485, PicoQuant), and fluorescence emission was collected by Hybrid Photomultiplier Detector Assembly 40 (PicoQuant) and processed by a TimeHarp 260 PICO Time-Correlated Single Photon Counting module (resolution 25 ps, PicoQuant). For each plant nucleus analyzed, at least 400 photons per pixel were recorded and fitted with bi-exponential decay function and convoluted with calculated instrument response function using SymPhoTime 64 software (PicoQuant).

## ABA uptake into yeast

Yeast cells were grown as mentioned above except that 50 ml of galactose-supplemented SD was used. The cells were grown in the presence or absence of 10 μM ABA, and as an additional control, a culture without ABA was treated with 10 μM ABA immediately prior to harvesting. Yeast cells were sedimented, washed three times with 25 ml of wash buffer (1 mM ammonium-acetate, 5 mM Tris–HCl, pH 5.8) and once with 5 mM EDTA (pH 8), and were resuspended in 500 μl methanol. Yeast cells were disrupted by glass beads as mentioned above and after removal of cell debris by centrifugation (20,000 *g*, 20 min, 4°C), the supernatant was evaporated and resuspended in 250 μl methanol/water (9:1; *v/v*). For ABA analysis, the samples were further diluted 1:5 (*v/v*) with the same solvent and subsequently infused (2 μl/min) into the mass spectrometer for Fourier-transform ion cyclotron resonance mass spectrometry (FT-ICR-MS). Direct-infusion FT-ICR mass spectra were acquired with a 12 T Bruker Solarix instrument (Bruker Daltonics, Bremen, Germany) equipped with an

APOLLO II electrospray ion source. Ion source settings were negative ionization, drying gas temperature 180°C, drying gas flow 4.0 l/min, and capillary voltage 3,600 V. Spectra were recorded with a time domain of four megawords, and 300 scans were accumulated for each experiment. ABA was quantified by addition of increasing amounts of ABA standard. Linear calibration models (*r* > 0.999) were computed from the peak areas of the ABA signals.

## Phosphoproteomic analysis

*Arabidopsis* 3xFLAG:SnRK2 proteins expressed in yeast were immunoprecipitated, tryptic digested, and subsequently analyzed for peptides of the activation loop. Briefly, yeast cells from 50 ml culture were harvested, washed with 10 mM EDTA (pH 8), and resuspended in 0.5 ml yeast lysis buffer (YLB; 50 mM Tris–HCl, pH 8; 1% DMSO; 100 mM NaCl, 1 mM EDTA; Halt™ Protease and Phosphatase Inhibitor Cocktail, Thermo Fisher Scientific). Cells were disrupted as mentioned above, and the cell-free protein extract was incubated with 20 μl ANTI-FLAG® M2 Affinity Gel (Sigma-Aldrich, Darmstadt, Germany) according to the manufacturer's instructions for 4 h. Bound proteins were washed three times with 1 ml buffer containing 50 mM Tris–HCl, 150 mM NaCl, pH 7.4, and proteins were solved from the affinity matrix into 20 μl sample buffer (Thermo Fisher scientific, Waltham, USA) at 70°C for 10 min prior to SDS gel electrophoresis, and Coomassie staining. Protein bands corresponding to the expected size of SnRK2 proteins were excised, reduced by DTT, and cysteine residues alkylated by chloroacetamide. In-gel digestion and peptide recovery were carried out as described (Shevchenko *et al*, 1996). Liquid chromatography-coupled mass spectrometry analysis was performed on a Q-Exactive HF and Orbitrap Fusion LUMOS (Thermo Fisher Scientific) coupled online to a Dionex 3000 HPLC system (Thermo Fisher Scientific). Samples were separated by a linear gradient for 60 min with PROCAL peptide standard added to a concentration of 100 fmol for each sample (Zolg *et al*, 2017). For phosphorylation site identification, all samples were measured on a Q-Exactive HF in data-dependent acquisition mode. Peptide and protein identification was performed using MaxQuant standard settings (version 1.5.8.3; Cox & Mann, 2008). Raw files were searched against the Araport11 database (https://www.araport.org/) for *Arabidopsis* and UniProtKB (https://www.uniprot.org/proteomes/UP000002311) for yeast proteins. Carbamidomethylated cysteine was set as fixed modification and phosphorylation of serine, threonine, or tyrosine, oxidation of methionine, and aminoterminal protein acetylation as variable modifications. Trypsin/P was specified as the proteolytic enzyme with up to two missed cleavages allowed. Results were adjusted to 1% false discovery rate for peptide spectrum matches, proteins, and phosphosite localization. To quantitate the phosphorylated peptides, a parallel reaction monitoring assay (PRM) was employed by constructing a spectral library for the selected peptides using the Skyline software (v4.1; MacLean *et al*, 2010) and a MaxQuant derived msms.txt file. Precursor charge states and transitions were chosen from the spectral library. In addition, the PROCAL retention time calibration mixture was monitored. PRM measurements were performed on an Orbitrap Fusion Lumos (Thermo Fisher Scientific) with the acquisition method switching between experiments after one duty cycle. The first experiment consisted of a full-scan MS1 spectrum recorded in the Orbitrap

(360–1,300 $m/z$, 15k resolution, AGC target 4e5, Maximum IT 50 ms). The second experiment consisted of a tMS2 PRM scan triggering MS2 scans based on an unscheduled list containing $m/z$ and charge information. For the tMS2 PRM scan, the scheduled precursors were isolated (isolation window 1.2 $m/z$), fragmented via HCD (NCE 30%), and recorded in the Orbitrap (120–2,000 $m/z$, 15k resolution, AGC target 2e5, Maximum IT 50 ms). RAW files were imported into Skyline for data filtering and analysis. Transition selection was manually refined to include site-determining ions for each phosphosite. The library ion match tolerance was set to 0.02 $m/z$, and transitions were extracted using the centroided product mass analyzer with 10 ppm mass accuracy and with high-selectivity extraction. Peaks were integrated using the automatic settings followed by manual curation of all peak boundaries and transitions. The data for summed area of fragment ion traces were exported for every transition, and the data were further analyzed in Microsoft Excel (version 2016). Phospho-peptide intensities were normalized to the corresponding unmodified peptide. Proteomic data can be downloaded from the ProteomeXchange consortium (ID: PXD013669) and the Panorama web repository server (https://panoramaweb.org/x8nkdT.url; ID: PXD013849).

**Data analysis**

Unless otherwise specified, all data were analyzed using Microsoft Excel 2016 or OriginPro 2018b. Outliers in Fig 2A were detected by Grubbs test (OriginPro 2018b; significance level 0.05). Significant difference in the mean was assessed by one-tailed Student's *t*-test for Fig 2A (unpaired, corrected for unequal variance, significance level 0.05) and two-tailed Student's *t*-test for Fig 6 (unpaired, corrected for unequal variance, significance level 0.05).

**Expanded View** for this article is available online.

## Acknowledgements

We thank Klaus Pflügler, Christoph Heidersberger, and Christian Kornbauer for technical assistance, and Farhah Assaad for comments on the manuscript. Special thanks to Gereon Czap for a first reporter construct and to Ramon A. Torres Ruiz and the CALM microscopy facility of the TUM for expertise and support. This work was supported by Deutsche Forschungsgemeinschaft GR938 and SFB924 "Molecular mechanisms regulating yield and yield stability in plants".

## Author contributions

EG supervised the project. MR and EG designed and coordinated the experiments with different parts of the research conceived by JK, PS-K, and BK. MR carried out the yeast experiments. JM and MR conducted the phosphosite analysis with contribution from KHE. DH and MR quantified ABA. SM performed the microscopic experiments, MP and SVT analyzed *Arabidopsis* protoplasts, and ID assayed *in vitro* phosphatase regulation. DC has been instrumental for implementation of the modular cloning system. All authors discussed and interpreted the data. MR and EG wrote the paper with contributions of all the authors. EG agrees to serve as the author responsible for contact and ensures communication.

## Conflict of interest

The authors declare that they have no conflict of interest.

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
