## [Review Process File · The EMBO Journal]

Rebuilding core abscisic acid signaling pathways of Arabidopsis in yeast

Moritz Ruschhaupt, Julia Mergner, Stefanie Mucha, Michael Papacek, Isabel Doch, Stefanie V. Tischer, Daniel Hemmler, David Chiasson, Kai H. Edel, Jörg Kudla, Philippe Schmitt-Kopplin, Bernhard Küster, and Erwin Grill

Review timeline:	Submission date:	21st Feb 2019
	Editorial Decision:	17th Apr 2019
	Revision received:	1st Jul 2019
	Accepted:	9th Jul 2019

Editor: Ieva Gailite

Transaction Report:

1st Editorial Decision

17th Apr 2019

Thank you for submitting your manuscript for consideration by the EMBO Journal. I sincerely apologise for the unusual delay in the assessment of your work due to belated submission of referee reports. We have now received two reports on your manuscript, which are included below for your information.

As you will see from the comments, both reviewers appreciate the work and the quality of the data and recommend publication of the manuscript. Given these positive evaluations from two experts of the field, I would like to invite you to submit a revised version of your manuscript, in which you address the comments of reviewer #2.

REFeree REPORTS:

Referee #1:

In general, a nice piece of work showing that yeast can be used to study the plant ABA signaling machinery. Grill's earlier discovery of the ABA receptor is perhaps one of the most important findings in plant basic research in the past two decades, up there with Estelle's discovery of the auxin receptor. The discovery that phosphatases are the direct target of the ABA receptor and that a kinase is also intimately involved are paradigms not seen in animal systems (yet), in terms of a protein phosphatase that is DIRECTLY regulated by a hormone receptor. Despite my great enthusiasm for Grill's earlier work and for the careful study described in this manuscript I am still left wondering whether the yeast system has really added a great deal of additional information not available with the protoplasts. However given the fact that yeast has not been used to reconstruct this machinery before and that without question it offers a great many new approaches to using synthetic biology for studying key aspects of the ABA dependent phosphatase and kinase, I highly recommend acceptance of this paper. This is not only excellent basic research but also, since ABA controls the movement of water out of a plant via stomata and also, so many other aspects of plant

life (embryo desiccation and then germination), that this will undoubtedly also prove to be a very important aspect of translational efforts for improving plant water use efficiency.

Referee #2:

This is a very interesting manuscript by Ruschhhaupt and colleagues. Here, the authors analyzed the core abscisic acid (ABA) signaling factors of *Arabidopsis thaliana* in yeast to dissect ligand-receptor specificities in a functional and multiplexed approach.

This study is very compelling and elegant. The authors took advantage of the yeast system to rapidly test many different combinations of pathway components and extract quantitative information. As mentioned in the manuscript, yeast was previously used successfully as a system to test many components of the auxin signaling pathways (Pierre-Jerome et al., PNAS 2014). In the present manuscript, Ruschhhaupt and colleagues go even deeper in the analysis of the ABA signaling core. They even identify SnRK2.1, 2.4, 2.5 and 2.10 (part of the subgroup I of the *Arabidopsis* SnRK2 protein kinase family) as putative candidates for being part of the ABA signaling core in *Arabidopsis thaliana*. The present manuscript should trigger new in planta studies, for example on roles of other SnRK2s in direct ABA signal transduction.

Overall, I believe the presented approach and results will be widely used and of broad interest to the research community.

Minor comments:

- 1) Line 135: the authors mention Fig. 1C but there is no such figure in the present manuscript.
- 2) Lines 148-149: RCAR1 does not activate signaling in yeast without exogenous ABA (Fig. 2A). However, this observation is true in *Arabidopsis* protoplasts (Fig. 2B).
- 3) Fig. 4C: is there any particular reason why OST1 was not used in this LUC-activity assay? OST1 is used throughout the manuscript for obvious reasons but not here. It should be included here as well.
- 4) Line 189: "subgroup II" is misspelled.

1st Revision - authors' response

1st Jul 2019

Referee #1:

In general, a nice piece of work showing that yeast can be used to study the plant ABA signaling machinery. Grill's earlier discovery of the ABA receptor is perhaps one of the most important findings in plant basic research in the past two decades, up there with Estelle's discovery of the auxin receptor. The discovery that phosphatases are the direct target of the ABA receptor and that a kinase is also intimately involved are paradigms not seen in animal systems (yet), in terms of a protein phosphatase that is DIRECTLY regulated by a hormone receptor. Despite my great enthusiasm for Grill's earlier work and for the careful study described in this manuscript I am still left wondering whether the yeast system has really added a great deal of additional information not available with the protoplasts. However, given the fact that yeast has not been used to reconstruct this machinery before and that without question it offers a great many new approaches to using synthetic biology for studying key aspects of the ABA dependent phosphatase and kinase, I highly recommend acceptance of this paper. This is not only excellent basic research but also, since ABA controls the movement of water out of a plant via stomata and also, so many other aspects of plant life (embryo desiccation and then germination), that this will undoubtedly also prove to be a very important aspect of translational efforts for improving plant water use efficiency.

Referee #2:

This is a very interesting manuscript by Ruschhhaupt and colleagues. Here, the authors analyzed the core abscisic acid (ABA) signaling factors of *Arabidopsis thaliana* in yeast to dissect ligand-receptor specificities in a functional and multiplexed approach.

This study is very compelling and elegant. The authors took advantage of the yeast system to rapidly test many different combinations of pathway components and extract quantitative information. As mentioned in the manuscript, yeast was previously used successfully as a system to test many components of the auxin signaling pathways (Pierre-Jerome et al., PNAS 2014). In the present manuscript, Ruschhaupt and colleagues go even deeper in the analysis of the ABA signaling core. They even identify SnRK2.1, 2.4, 2.5 and 2.10 (part of the subgroup I of the Arabidopsis SnRK2 protein kinase family) as putative candidates for being part of the ABA signaling core in Arabidopsis thaliana. The present manuscript should trigger new in planta studies, for example on roles of other SnRK2s in direct ABA signal transduction. Overall, I believe the presented approach and results will be widely used and of broad interest to the research community.

We very much appreciate the input of the editor and both reviewers, and their positive feedback. We addressed the points raised one-by-one as stated below.

Minor comments:

1) Line 135: the authors mention Fig. 1C but there is no such figure in the present manuscript.

Response: Sorry, meant is Fig. 1B.

2) Lines 148-149: RCAR1 does not activate signaling in yeast without exogenous ABA (Fig. 2A). However, this observation is true in Arabidopsis protoplasts (Fig. 2B).

Response: There is an approximately twofold induction of ABA signaling by RCAR1 in yeast ($p < 0.001$; one-tailed t-test; see also now Table 2 of the appendix).

3) Fig. 4C: is there any particular reason why OST1 was not used in this LUC-activity assay? OST1 is used throughout the manuscript for obvious reasons but not here. It should be included here as well.

Response: The data for OST1 have been presented in Fig. 1A and was therefore omitted. We include now the data for better comparison in Fig. 4C.

4) Line 189: "subgroup II" is misspelled.

Response: corrected.

2nd Editorial Decision

9th Jul 2019

Thank you for submitting the revised version of your manuscript. The main issues have now been addressed and I am happy to inform you that your manuscript has been accepted for publication. Congratulations on a nice study!

Corresponding Author Name: Erwin Grill

Journal Submitted to: EMBO J

Manuscript Number: 101859